



# Distribution and morphology of non-persistent and persistent contrail formation areas in ERA5.

Kevin Wolf[1,a], Nicolas Bellouin[1,2], and Olivier Boucher[1]

[1]Institut Pierre-Simon Laplace, Sorbonne Université / CNRS, Paris, France
[2]Department of Meteorology, University of Reading, Reading, United Kingdom
[a]now at Leipzig Institute for Meteorology (LIM), Leipzig University, Leipzig, Germany

**Correspondence:** Kevin Wolf (kevin.wolf@uni-leipzig.de)

**Abstract.** The contrail formation potential as well as its temporal and spatial distribution are estimated using meteorological conditions of temperature and relative humidity from the ERA5 re-analysis provided by the European Centre for Medium-Range Weather Forecasts. Contrail formation is estimated with the Schmidt–Appleman criterion (SAc), solely considering thermodynamic effects. The focus is on a region ranging from Eastern United States to central Europe. Around 18,000 flight trajectories from the In-service Aircraft for a Global Observing System (IAGOS) are used as a representative subset of transatlantic, commercial flights. The typical crossing distance through a contrail-prone area is determined based on IAGOS measurements of temperature T and relative humidity r, then based on co-located ERA5 simulations of the same quantities. For IAGOS, 50 % of the crossings of persistent contrail (PC) regions are shorter than 9 km, while in ERA5 the median is 155 km. Time-averaged IAGOS data lead to a median crossing length of 66 km. The difference between the two data sets are attributed to the higher variability of $r$ in IAGOS compared to ERA5. Binary masks of PC formation are created by applying the SAc on the two-dimensional fields of $T$ and $r$ from ERA5. In a second step the morphology of PC regions is also assessed. Half of the PC regions are found to be smaller than $\approx$35000 km$^2$ (at 200 hPa) and the median of the maximum dimension is shorter than 760 km (at 200 hPa). Furthermore, PC regions tend to be of near-circular shape with a tendency to a slight oval shape and a preferred alignment along the dominant westerly flow. Seasonal, vertical distributions of PC formation potential $\mathcal{P}$ are characterized by a maximum between 250 and 200 hPa. $\mathcal{P}$ is subject to seasonal variations with a maximum in magnitude and extension during the winter months and a minimum during summer. The horizontal distribution of PC regions suggests that PC regions are likely to appear in the same location on adjacent pressure levels. Climatologies of $T$, $r$, wind speed $U$, and resulting PC formation potential are calculated to identify the constraining effects of $T$ and $r$ on $\mathcal{P}$. PC formation is primarily limited by too warm conditions below and too dry conditions above the formation region. The distribution of PCs is slanted towards lower altitudes from 30°N to 70°N, following lines of constant $T$ and $r$. For an observed co-location of high $U$ and $\mathcal{P}$ it remains unclear whether PC formation and the jet stream are favored by the same meteorological conditions or if the jet stream itself favors PC occurrence. This analysis suggests that some PC regions will be difficult to avoid by rerouting aircraft because of their large vertical and horizontal extents.



## 1 Introduction

Aviation is a contributor to global climate warming by being responsible for 2.5 to 2.6 % of the global carbon dioxide ($CO_2$) emissions in 2018 (Friedlingstein et al., 2019; Lee et al., 2021; Boucher et al., 2021). However, $CO_2$ is only partly responsible for the climatic effect of aviation. Further contributions to aviation-induced climate change stem from the by-products of fossil fuel combustion like nitrogen oxides ($NO_x$) or sulfur dioxide ($SO_2$). Additionally, burning fuels that contain hydrogen bonds, no matter whether of fossil or synthetic origin, will result in the emission of water vapor (WV), which is contained in the exhaust. Subsequently, the exhaust WV can condense and lead to the formation of condensation trails, also termed contrails (Schumann, 1996; Kärcher, 2018). Such contrails are artificial, optically thin cirrus-like clouds that are known to have, on average, a net warming effect on global climate (Burkhardt and Kärcher, 2011; Schumann et al., 2015; Lee et al., 2021).

Typically, the cooling or warming of an atmospheric perturbation, here contrails, is quantified by its radiative forcing (RF). The RF is defined as the difference in the net irradiance at the top-of-atmosphere (TOA) with and without a perturbation. For $CO_2$, for example, Lee et al. (2021) and Boucher et al. (2021) estimated an aviation-related RF of around 30 $mWm^{-2}$. While this $CO_2$-related RF is relatively certain, the RF of WV, contrails, and induced cirrus is assumed to be at least similar or even higher than that of $CO_2$ but subject to large uncertainties (Burkhardt and Kärcher, 2011; Lee et al., 2021).

Due to their warming effect, contrail avoidance and mitigation has seen a growing interest in recent years. Whether a contrail forms or not can be estimated with the Schmidt–Appleman criterion (SAc; Schmidt, 1941; Appleman, 1953). The SAc is based solely on thermodynamic principles and provides critical thresholds of temperature and relative humidity beyond which a contrail can form. Favorable formation conditions occur, when the ambient air is colder than a critical temperature $T_{crit}$ and the ambient air is moister than a critical relative humidity $r_{crit}$. For a contrail to be persistent, the ambient air must additionally be ice supersaturated (ISS) in so called ice supersaturated regions (ISSRs).

Regions that are prone to persistent contrail (PC) formation (life time > 10 min) that exert a net warming might be actively avoided on a case-by-case basis by re-routing individual flights. Active re-routing relies on numerical weather prediction models (NWP) and subsequent estimations of ISSR occurrence and contrail RF. However, this requires the accurate prediction of ISSR in space and time and, more importantly, a flexible flight planning and dispatch (Williams and Noland, 2005; Schumann et al., 2011; Irvine et al., 2014; Teoh et al., 2020).

In a previous study, Wolf et al. (2023a) used radiosonde observations to investigate the temporal and spatial distribution of the contrail formation potential. The study was limited to one single station close to Paris, France, which limits the spatial representativeness. Therefore, a spatially extended data set and contrail statistics are required (Gierens and Spichtinger, 2000).

Reliable and homogeneously distributed observations of temperature and relative humidity at flight altitude, i.e., above 30,000 ft ($\approx$ 10 km) are sparse. A complement to in situ observations is the High Resolution component (HRES) of ERA5 (Hersbach et al., 2020) provided by the European Centre for Medium-Range Weather Forecasts (ECMWF). Similar to NWP, using the ERA5 reanalysis model to estimate contrail occurrence, one relies on accurate data assimilation of the sparse in situ





observations, assimilated satellite observations, and the correct representation of temperature, relative humidity, and resulting ISS. ERA5 skill at simulating ISS has been assessed against in situ observations, e.g., by Wolf et al. (2023b). One extensive data set that is available in this regard is the In-service Aircraft for a Global Observing System (IAGOS; Petzold et al.,
2015). IAGOS is a network of commercial aircraft performing in situ measurements of meteorological conditions, trace gas concentrations, and cloud properties.

Wolf et al. (2023b) used IAGOS observations to validate ERA5 performance in terms of temperature, relative humidity, and contrail formation potential for $p$-levels between 250 and 175 hPa. In general, a good agreement in temperature was found, while larger differences were identified for relative humidity. Similar results, indicating a dry bias of ERA5 in the
upper troposphere, were identified, e.g., by Kunz et al. (2014), Dyroff et al. (2015), Gierens et al. (2020), Bland et al. (2021), and Schumann et al. (2021). As a consequence of the underestimation of ISS, a slight underestimation of PC occurrence was identified (Wolf et al., 2023b). To overcome the dry bias, Wolf et al. (2023b) proposed and applied a bias correction technique based on quantile mapping (QM). Using the QM-method and removing the dry bias from the ERA5 data, the effect on estimated ISS and PC formation was found to be minor. Therefore, it was argued that ERA5 performs well in terms of the
statistical representation of ISSR and PC occurrence.

In Wolf et al. (2023b) the QM-technique and model validation were centered to a region spanning from 110°W to 30°E and 30°N to 70°N covering the majority of the air traffic between eastern United States and central Europe, i.e., along the North Atlantic Flight Tracks, officially titled the North Atlantic Organized Track System (NAT-OTS). Historically, this is also the region with the most frequent and dense IAGOS observations (Petzold et al., 2020). Due to the concentrated air traffic in this
region and the confidence in ERA5 in terms of PC representation, the present study focuses on the same domain.

Within the present study, we provide distributions of PC crossing distance using ERA5 and IAGOS observations. Furthermore, the morphology of PC formation regions in terms of size, orientation, major axis length, and aspect ratio of individual regions are presented. Information about the shape is important for economic decision making to re-route flights horizontally or vertically. Also seasonal, vertical distributions of the PC formation potential are calculated that could be used by airlines to
assess the distance they would need to reroute on average. In this context, the potential for overlapping PC formation regions of adjacent PC formation layers is investigated. The overlap potential is relevant to vertical re-routing. Finally, climatologies of temperature, relative humidity, wind speed, and related PC formation potential are presented. These climatologies provide a general perspective of the temporal and spatial distribution of PC regions in ERA5.

This introduction is followed by Section 2 that briefly outlines the utilized IAGOS data set and the ERA5 model data.
Furthermore, the basics of the SAc are explained and the methods that are used to determine the PC morphology. In Section 3 the results are discussed, which are then summarized and discussed in Section 4.



## 2 Data and Methods

### 2.1 In-service Aircraft for a Global Observing System

In situ observations are obtained from the In-service Aircraft for a Global Observing System (IAGOS; Petzold et al., 2015).

IAGOS is supported by commercial airlines, which provide a part of their fleet as a platform for scientific measurements. Selected aircraft are equipped with sensors to measure meteorological conditions, trace gas concentrations, and cloud properties. Since 2015, all aircraft within the IAGOS network are equipped with the 'Package 1' (P1) instrument package system. The P1 package includes, among others, a separate sensor package 'ICH' that measures temperature $T_{\mathrm{P1}}$ (PT-100 platinum sensor) and relative humidity $r_{\mathrm{P1}}$. $r_{\mathrm{P1}}$ (defined with respect to liquid water) is measured by a capacitive sensor (Humicap-H, Vaisala,

Finland). Both sensors are mounted to the aircraft fuselage in a Model 102 BX housing of Rosemount Inc. (Aerospace Division, USA), which minimizes solar heating and thermodynamic effects. The obtained raw data is post-processed by the IAGOS consortium according to Helten et al. (1998) and Boulanger et al. (2018, 2020). During the post-processing an "in-flight calibration method" (IFC) is applied that corrects offset drifts that might have occurred during the course of the deployment period (Smit et al., 2008; Petzold et al., 2017).

The IAGOS post-processed data of temperature $T_{\mathrm{P1}}$ and relative humidity $r_{\mathrm{P1}}$ are published with a temporal resolution of 4 s. However, the response time $t_{1-1/\mathrm{e}}$ is a critical sensor characteristic that has to be considered. $t_{1-1/\mathrm{e}}$ is typically defined as time that is required by a sensor to adapt to $1 - \frac{1}{e} = 0.6\bar{3}$ of a sudden change in the measured quantity. For the temperature sensor a response time $t_{1-1/\mathrm{e}}$ of 4 s is reported. Due to the measurement principle of the relative humidity sensor, the humidity sensor has a response time $t_{1-1/\mathrm{e}}$ that is temperature dependent. For temperatures around 293 K, $t_{1-1/\mathrm{e}}$ is in

the range of 1 s. When the temperature approaches 233 K $t_{1-1/\mathrm{e}}$ increases up to 180 s. The reason for the increase in $t_{1-1/\mathrm{e}}$ is the reduced molecular diffusion of water vapor into and out of the sensors polymer substrate. Consequently, for conditions with temperatures $T$ = 293 K, the distance between two IAGOS measurements of $T_{\mathrm{P1}}$ and $r_{\mathrm{P1}}$ is 0.96 km at a cruise speed of 240 m s$^{-1}$, while the increase in $t_{1-1/\mathrm{e}}$ leads to an average over a distance between 15 km (253 K) and up to 50 km (233 K) at cruise altitude. For the temperature sensor an accuracy of $\pm 0.5$ K is reported and the relative humidity sensor is

characterized by an average uncertainty of $\pm 6$ %. Considering sensor calibration and data post-processing the total uncertainty in $r$ is estimated to be between 5 % and up to 10 %, generally increasing with decreasing temperature (Helten et al., 1998).

The available IAGOS measurements are filtered for data quality and are limited to the domain of interest. In this study, only measurements that pass the following criteria are used:

– IAGOS quality flag of $T_{\mathrm{P1}}$ and $r_{\mathrm{P1}}$ is 'good' and 'limited'

– measurements are located between 30°N and 70°N

– measurements are between 325 and 150 hPa

– $r_{\mathrm{P1}}$ (w.r.t liquid water) is between 0 and 100 %



A density map of the measurements from the filtered flights can be found in Wolf et al. (2023b). Furthermore, we use the IAGOS observations as a proxy for commercial air traffic and derive flight pressure distributions (FPDs) as well as flight latitude distributions (FLatDs) for the entirety of the three sub-domains: Eastern United States (US, 110°W–65°W), the North Atlantic (NA, 65°W–5°W), and Europe (EU, 5°W–30°E). It is noted that only IAGOS-contributing aircraft are included in our statistics, which represent a very small fraction of the total flight traffic. But IAGOS flight should be representative of where and when commercial aircraft fly over the North Atlantic.

## 2.2 ERA5

Meteorological data are downloaded from the ECMWF Copernicus Climate Data Store (Hersbach et al., 2023). More specifically, we use the High Resolution component (HRES) of ERA5 (Hersbach et al., 2020) with the maximal spatial and temporal resolution of $0.25° \times 0.25°$ and one hour, respectively. The ERA5 data set was generated with the ECMWF Integrated Forecasting System (IFS) cycle Cy41r2 (operational in 2016). Actual IAGOS flight trajectories are used to extract along-track temperature $T_{\mathrm{ERA}}$, relative humidity $r_{\mathrm{ERA}}$, and wind speed $U_{\mathrm{ERA}}$. The variables are extracted by selecting the temporally and spatially closest (nearest neighbor) ERA5 grid point with respect to the IAGOS observations. Temporal and spatial interpolation is avoided as relative humidity is sensitive to the applied interpolation technique in time and space (Schumann, 2012).

Depending on $T_{\mathrm{ERA}}$ of the grid box, the relative humidity $r_{\mathrm{ERA}}$ is provided with respect to liquid water or ice. To be consistent and to apply the SAc on the extracted ERA5 data, all values of $r_{\mathrm{ERA}}$ are converted to either be defined over liquid water or ice. Details and the equations that are used for the conversion are given in Wolf et al. (2023b) Sec. 2.2 ERA5. Subsequently, the converted values are referred to as $r_{\mathrm{ERA}}$ (w.r.t. liquid water) and $r_{\mathrm{ERA,ice}}$ (w.r.t. ice). Similarly, relative humidity from IAGOS is labeled as $r_{\mathrm{P1}}$ (liquid water) and $r_{\mathrm{P1,ice}}$ (ice).

Due to the fixed grid resolution fixed grid resolution of $0.25°$ in ERA5, the distance between two points on same lines of longitude depends on the latitude. With the focus of this study on the domain between 30°N and 70°N, the distance between adjacent points along the longitude ranges between 24 km at 30°N and 14 km at 70°N. For simplicity we use an average grid box size of 19 km. While the IAGOS observations are recorded every 4 s and the relative humidity measurements are already averaged by the sensor time lag, IAGOS $r_{\mathrm{ice}}$ is additionally averaged to bridge the difference in the spatial resolution of ERA5 and IAGOS. We apply a running average of 19·4 s as in Wolf et al. (2023b) and refer to this product as IAGOS time-averaged.

IAGOS data are mapped onto certain pressure ($p$)-levels from ERA5. The assignment is realized by pressure brackets that enclose the ERA5 $p$-levels. $p$-levels and the associated pressure ranges are given in Table 1.

## 2.3 Schmidt-Appleman criterion, potential contrail formation, and contrail persistence

Contrails form only under certain ambient conditions. For contrail formation to take place, the surrounding air must be bellow a critical temperature $T_{\mathrm{crit}}$ and above a critical relative humidity $r_{\mathrm{crit}}$. These thresholds are commonly determined by the Schmidt-Appleman criterion (SAc, Appleman, 1953; Schumann, 1996). The SAc is a first-order approximation as it only considers thermodynamic principles but neglects potential dynamical effects that take place in the vortex behind the aircraft.



**Table 1.** ERA5 pressure levels (in unit of hPa) and pressure ranges used to collocate the IAGOS observations.

| Pressure level (hPa) | Pressure range (hPa) |
|---|---|
| 300 | $275.0 \leq p < 325.0$ |
| 250 | $237.5 \leq p < 275.0$ |
| 225 | $212.5 \leq p < 237.5$ |
| 200 | $187.5 \leq p < 212.5$ |
| 175 | $150.0 \leq p < 187.5$ |

The SAc allows to estimate general contrail formation (necessary criterion) but it is insufficient to identify contrail persistence. Persistent contrails (life time > 10 min) additionally require ISS of the ambient air with respect to ice ($r_{\text{ice}} > 100\,\%$).

Within this study we use the revised version of the SAc following Schumann (1996) and Rap et al. (2010). General details on the SAc and equations required to calculate $T_{\text{crit}}$ and $r_{\text{crit}}$ can be found in Rap et al. (2010) or Wolf et al. (2023a). Within the present study the same definitions and nomenclature as in Wolf et al. (2023a) are used, and data points are categorized for non-persistent contrails (NPC), persistent contrails (PC), and reservoir (R) conditions. Details for the group of Reservoir conditions that are supersaturated but not fulfilling the SAc can be found in Wolf et al. (2023a). All data points that are not assigned to one of the groups are labeled as non-contrail formation (NoC).

$T_{\text{crit}}$ and $r_{\text{crit}}$ from the SAc are fuel dependent. The focus of this paper is on the effects of contrails that form by burning Jet-A1, also known as kerosene. Therefore, values of the specific heat capacity $Q = 43.2\,\mathrm{MJ\,kg^{-1}}$ and water-vapor-emission–index $EI = 1.25\,\mathrm{kg\,kg^{-1}}$ are used in the SAc (Schumann, 1996). In addition, the engine efficiency $\eta$ is set to 0.3 (Rap et al., 2010).

## 2.4 Estimation of morphology of persistent contrail regions

Applying the SAc, as described in Section 2.3, each grid box in the ERA5 4D data set is classified as 'NPC', 'PC', 'R', and 'NoC'. Masking ERA5 data according to the PC formation flags, creates binary images that can be processed with the python `scikit-image` package, which was originally developed for image processing (van der Walt et al., 2014). The package allows to identify and label features in images, here the individual regions of PC formation in our case. The class `skimage.measure.regionprops` includes the functions `area`, `axis_minor_length`, `axis_major_length`, and `orientation` that return the total number of adjacent grid-boxes, the length of the minor and major axis of the PC structure, and the orientation of the individual identified regions, respectively. The aspect ratio $\mathcal{Z}$ is calculated by dividing the minor axis length by the major axis length. The major axis length is converted into the maximum dimension $D$ (in unit of kilometers) by multiplying the major axis length by two and the grid-box mean edge length of 19 km (grid box resolution at 50°N). Similarly, the area $\mathcal{A}$ of a PC region is calculated by multiplying the number of grid-boxes returned by the function `area` with a factor of $19 \cdot 19\,\mathrm{km^2} = 361\,\mathrm{km^2}$. The function `orientation` returns the angle $\gamma$ between the major axis length and the rows in the matrix, in this case parallels. Therefore, an angle of $\gamma = 0°$ indicates a PC region with $D$ along a parallel,



while an angle of $\gamma = 90°$ indicates a PC region with $D$ along a meridian. For almost circular PC regions, where $\mathcal{Z} > 0.95$, the orientation cannot be determined accurately and the values are excluded from the calculation of the orientation. PC regions that touch the boundaries of the domain are kept in the analysis but are flagged. In that way edge-touching PC regions, mostly the largest ones, are retained in the analysis, while allowing to quantify the impact of the boundary-interacting PC regions

on the calculations. PC regions that touch the boundaries are assumed to be larger than determined by the routine as some parts go beyond the defined region. It is noted that although the detection of PC formation regions with the `scikit-image` package is straightforward it is however difficult to explicitly assign small, individual PC regions to a larger group that might be considered as one PC region. The definition of a larger cluster would then depend on the allowed distance between PC regions. Therefore, each individual PC region is treated separately even though they might belong to a larger PC cluster.

## 3 Results

### 3.1 Characteristic crossing length of contrail formation regions

IAGOS observations are exploited to estimate the characteristic crossing length $\mathcal{L}$, which can be understood as the distance an aircraft flies within (crosses) a region that allows for NPC or PC formation. Values of $\mathcal{L}$ are calculated on the native temporal and spatial resolution of IAGOS with 4 s sampling. It is recalled that the response time $t_{1-1/e}$ of the relative humidity

sensor and associated time-averaging leads to an averaging over 15 to 50 km at flight altitude, which is in the range of the 19 km (50°N) grid-box. Smoothing over longer periods of time and distances shifts the distribution of $\mathcal{L}$ to larger values. $\mathcal{L}$ is determined by counting the number of consecutive along-track IAGOS or ERA5 segments that were flagged for NPC and PC conditions. The algorithm prioritizes PC samples over NPC formation, as the criteria for PC (SAc plus ISS) are stricter than for NPC. Furthermore, we assume that PC are embedded within NPC regions as the transition from PC to NPC domains follows

a continuous decrease in $r_{ice}$ from within the PC center to the edge of the NPC region. Consequently, NPC measurements are considered as consecutive even when they are interrupted by PC flagged measurements. In all other cases, a series of consecutive flags is interrupted when at least two consecutive samples belong to another category. Note that the estimated $\mathcal{L}$ depends on how the contrail formation regions are oriented with respect to the flight track. Therefore, $\mathcal{L}$ is always smaller than the maximum dimension of the contrail formation region (Dorff et al., 2022). However, the knowledge of typical values for $\mathcal{L}$

is an important information in the decision making to cross or avoid persistent contrail formation regions.

Cumulative distribution functions (CDFs) of $\mathcal{L}$ for NPC and PC regions are presented in Fig. 1a and b, respectively. In general, distributions from the native IAGOS resolution (black line) are steepest, which indicates that the distributions are mostly dominated by short $\mathcal{L}$. The majority of crossing lengths, given by the 75th percentile, are below 70 and 92 km for PC and NPC, respectively. This is equivalent to a flight time of around 5 to 7 min at 800 km h$^{-1}$. Half of the crossing length (50th

percentile) are shorter than 9 and 12 km for PC and NPC (1 min at 800 km h$^{-1}$), respectively. Towards $\mathcal{L} > 800$ km the CDF reaches an asymptote.

Calculated distributions of $\mathcal{L}$ on the basis of ERA5 along-track samples lack the shortest $\mathcal{L}$. Consequently, the fraction of larger $\mathcal{L}$ is enhanced compared to IAGOS. Half of the crossing lengths, given by the 50th percentile, are shorter than 155 km



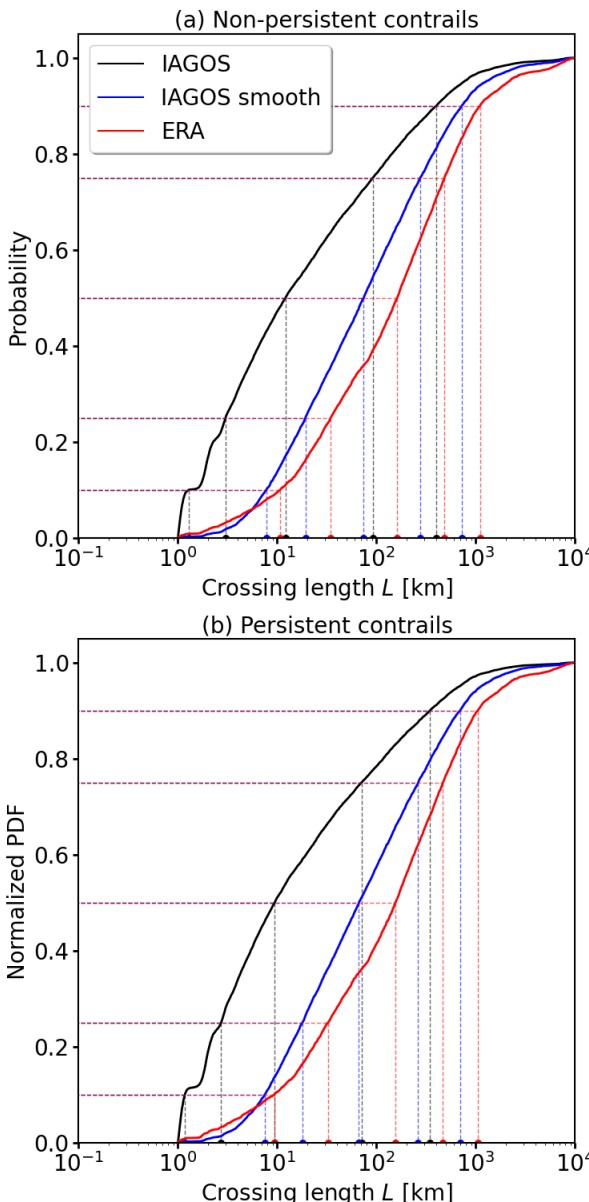

**Figure 1.** Cumulative distribution functions (CDFs) of the characteristic crossing length $\mathcal{L}$ (in unit of km) based on individual transects of aircraft passing through **(a)** non-persistent contrail (NPC) or **(b)** persistent contrail (PC) formation regions. $\mathcal{L}$ inferred from native IAGOS observations and time-averaged IAGOS observations are given in black and blue, respectively. $\mathcal{L}$ from ERA5 along-track data are given in red. $10^{\text{th}}$, $25^{\text{th}}$, $50^{\text{th}}$, $75^{\text{th}}$, and $90^{\text{th}}$ percentile are indicated by dashed lines.

for PC and 161 km for NPC, both equivalent to 12 min flight time at $800\,\text{km}\,\text{h}^{-1}$. This is approximately a factor of 15 larger
compared to $\mathcal{L}$ determined from IAGOS (native resolution). The differences in the distribution of $\mathcal{L}$ are attributed to the spatial



**Table 2.** Length of flight transects $\mathcal{L}$ through regions of non-persistent (NPC) and persistent (PC) contrail formation. $\mathcal{L}$ are given for the 5, 10, 25, 50, and 75<sup>th</sup> percentiles ($Q$).

| Condition | Crossing length $\mathcal{L}$ [km] | | |
|---|---|---|---|
| | IAGOS | IAGOS (time-averaged) | ERA5 |
| NPC (10<sup>th</sup>-%) | 1 | 8 | 11 |
| NPC (25<sup>th</sup>-%) | 3 | 19 | 35 |
| NPC (50<sup>th</sup>-%) | 12 | 74 | 161 |
| NPC (75<sup>th</sup>-%) | 92 | 274 | 482 |
| NPC (90<sup>th</sup>-%) | 399 | 720 | 1100 |
| PC (10<sup>th</sup>-%) | 1 | 6 | 9 |
| PC (25<sup>th</sup>-%) | 3 | 18 | 32 |
| PC (50<sup>th</sup>-%) | 9 | 66 | 155 |
| PC (75<sup>th</sup>-%) | 70 | 262 | 465 |
| PC (90<sup>th</sup>-%) | 346 | 692 | 1043 |

resolution of ERA5, where short crossing lengths occur less frequently and cannot by construction be smaller than grid-box size.

In a previous study that estimated $\mathcal{L}$ from IAGOS observations, Wilhelm et al. (2022) identified similar $\mathcal{L}$ for PC regions, with the majority (87 %) of the analyzed 'Big Hits' flights being shorter than 75 km. The difference to our estimates emerge

from the two different approaches applied here. While we used the contrail potential from the SAc, Wilhelm et al. (2022) considered only contrails that additionally exert an instantaneous radiative effect larger than 19 W m$^{-2}$ which they consider as 'Big Hits'.

Even though the IAGOS measurements are averaged due to the response time of the relative humidity sensor, the difference in the spatial resolution of IAGOS and ERA5 propagates in the distributions of $\mathcal{L}$. To better estimate the impact of the spatial

resolution, $\mathcal{L}$ for NPC and PC is determined on basis of the time-averaged IAGOS observations. Calculated CDFs of $\mathcal{L}$ for the time-averaged IAGOS data set (Fig. 1; blue lines) show a better agreement with the ERA5-based distributions, particularly for $\mathcal{L} < 10$ km. For $\mathcal{L} > 10$ km the discrepancies between time-averaged IAGOS data and ERA5 increase (please note the $x$-axis log scale). However, the time-averaged data better represents the distribution of $\mathcal{L}$ that is obtained from ERA5. After the averaging, half of the crossing lengths (50<sup>th</sup> percentile) are shorter than 66 and 74 km for PC and NPC, respectively.

For the same probability, $\mathcal{L}$ from time-averaged IAGOS data are approximately a factor of 7 larger than the native IAGOS observations and a factor of two smaller than $\mathcal{L}$ determined from ERA5. In spite of the relative humidity sensors inertia and additional smoothing, a mismatch with respect to the ERA5 distributions remains, which can be attributed to a smaller amount of variability in $r$ in ERA5 compared to IAGOS. A summary of the results of $\mathcal{L}$ is given in Table 2.

In theory, the maximum detectable $\mathcal{L}$ using IAGOS is limited by the longest flight in the data set. Similarly, a lower boundary

of $\mathcal{L}$ exists, which is limited by the response time $t_{1-1/e}$ of the relative humidity sensor and, hence, the ability to detect small-





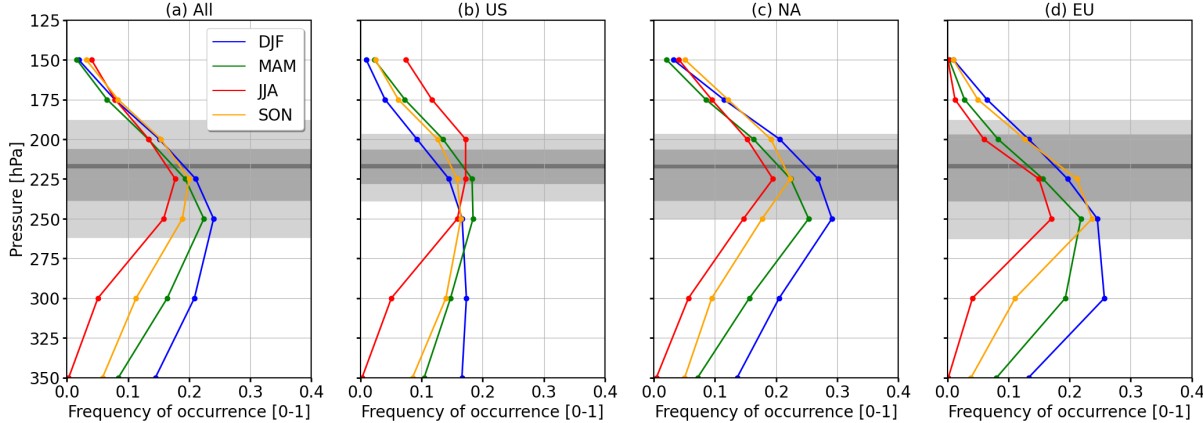

**Figure 2.** Vertical distribution of persistent contrail formation potential $\mathcal{P}$ (unitless) for the **(a)** full domain, as well as **(b)** the US, **(c)** the North Atlantic Ocean, and **(d)** the European sub-domains, respectively. Colors represent winter (blue), spring (green), summer (red), and autumn (orange). IAGOS-based flight pressure distributions are indicated in gray giving the $10^{th}$, $25^{th}$, $50^{th}$, $75^{th}$, and $90^{th}$ percentiles.

scale fluctuations in the relative humidity field. It is also hypothesized that $\mathcal{L}$ has a natural lower boundary between 5 and 10 km (Spichtinger and Leschner, 2016), with the potential explanation that meso-scale turbulence and mixing processes stratify the humidity distribution (Diao et al., 2014). However, the existence of such a lower, hypothetical boundary of $\mathcal{L}$ has not been explicitly formalized yet.

Instruments that better resolve relative humidity in time do exist. For example, Diao et al. (2014) used airborne measurements of an open-path vertical-cavity surface-emitting laser (VCSEL) hygrometer (Zondlo et al., 2010) to estimate the crossing length $\mathcal{L}$ across ISSR. They found mean and median $\mathcal{L}$ of 3.5 and 0.7 km, respectively, which is two orders of magnitude smaller than what we derived. They further noted that this is two orders of magnitude lower compared to other studies before, for example by Gierens and Spichtinger (2000), who identified mean $\mathcal{L}$ of ISSR of 150 km on basis of IAGOS flights. It is also

highlighted that Diao et al. (2014) and Gierens and Spichtinger (2000) investigated $\mathcal{L}$ of ISSR, while we estimate $\mathcal{L}$ for NPC and PC regions, which also consider the SAc.

### 3.2    Vertical distribution of persistent contrail formation potential

We now derive the vertical distribution and the vertical extent of PC regions in the ERA5 data set. Regional variations, i.e., longitudinal dependencies, are considered by sub-sampling the full domain for the Eastern United States (US, 110°W–65°W),

the North Atlantic (NA, 65°W–5°W), and Europe (EU, 5°W–30°E). Subsequently, the focus is on PC formation on $p$-levels 250, 225, and 200 hPa. The vertical distributions of PC occurrence are expressed as the PC formation potential $\mathcal{P}$ (unitless), which is shown in Fig. 2 for the different regions and seasons. $\mathcal{P}$ is calculated for each $p$-level as the ratio of PC flagged grid-boxes in relation to all grid boxes, and is then averaged over time steps and months.



First, we consider $\mathcal{P}$ of the full domain (Fig. 2a). Above 225 hPa, $\mathcal{P}$ is characterized by a small seasonal variability. A
maximum $\mathcal{P}$ is identified on $p$-level 250 hPa with 0.24 (winter). For altitudes below 225 hPa the distributions are dispersed
suggesting a larger seasonal variability. Considering only the most frequented pressure levels (gray areas), $\mathcal{P}$ is generally lowest
in summer with a minimum of 0.13 at 200 hPa, while higher $\mathcal{P}$ values are found in winter with a maximum of 0.24 at 250 hPa.
Spring and autumn lie between those two extremes. Such a seasonal pattern is consistent with earlier observations of ISSR
occurrence and PC formation as reported, e.g., by radiosonde-based studies by Spichtinger et al. (2003) or Wolf et al. (2023a).
Figure 2a further suggests that maximum $\mathcal{P}$ overlaps with the most frequented flight levels. Hence, the majority of commercial
aircraft are currently flying at altitudes that are most prone to PC formation. Considering the full domain, shifting flights to
higher altitudes would reduce the chance to form PC.

Narrowing down on the regional aspect of PC occurrence, similar distributions are found with only small seasonality above
200 hPa (see Fig. 2b–d). An exception is the US domain, revealing particularly high $\mathcal{P}$ during summer. Furthermore, the order
of $\mathcal{P}$ is reversed in relation to the other sub-domains, with the maximum of 0.18 (200 hPa) in summer and lowest $\mathcal{P}$ of 0.09 in
winter (200 hPa). Within the NA sub-domain, a maximum of $\mathcal{P} = 0.28$ at 250 hPa is found in winter and a minimum $\mathcal{P} = 0.15$
at 250 hPa appears in summer. For the EU sub-domain minimal $\mathcal{P}$ appear in summer with 0.08 at 200 hPa, and similar maxima
in winter and spring with $\mathcal{P} = 0.24$ at 250 hPa.

The reordering of $\mathcal{P}$ in the US domain and the shift of maximal $\mathcal{P}$ from higher to lower altitudes, when moving from
west to east, are intriguing features. These two patterns might be explained by the general circulation, the typical location
of the jet stream, and the topography of the North American continent. The topography of North America allows cold and
dry Arctic air masses to reach far south. Smith and Sheridan (2020) reported that large-scale cold air outbreaks occur more
often over the North American continent than over the mid-Atlantic and Europe throughout the year. While such conditions
with low temperature are required for PC formation, the lack of humidity inhibits PC formation. Also contrails appear more
frequent in coastal regions of the North American continent, where humidity from the Pacific Ocean and the Gulf of Mexico
provide humidity for contrail formation (Avila et al., 2019). The generally higher $\mathcal{P}$ over the North Atlantic and EU domain
are explained by the frequent influence of the jet stream, which controls storm formation and the location of the North Atlantic
storm tracks. Cyclonic activity and atmospheric rivers / warm conveyor belts advect humidity from the surface level, which
is lifted and cooled, and favors contrail formation (Gettelman et al., 2006a, b; Dacre et al., 2015; Spichtinger and Leschner,
2016). With storm activity being lower during summer and increased in winter (Eiras-Barca et al., 2016), humidity advection is
intensified during the winter months. This matches with highest $\mathcal{P}$ in winter and it explains the seasonality in the distributions
of $\mathcal{P}$ for the NA and EU domains.

Figure 2 suggests that $\mathcal{P}$ is a continuous function of $p$. Hence, adjacent $p$-levels, for a given time step, might be equally prone
to PC formation and contrail avoidance by vertical rerouting might be impractical. Therefore, we estimate the frequency of the
vertical fractional overlap or vertical intersection $\mathcal{I}$ (unitless) of PC regions between adjacent $p$-levels. $\mathcal{I}$ can be interpreted
as an indicator of the vertical extension or cohesion of PC regions, and provides information about how likely it is that two
adjacent layers allow PC formation. The intersection of adjacent $p$-levels is determined with the binary flag of PC occurrence
(PC can form = 1, no PC formation = 0). Binary multiplication of adjacent $p$-levels for individual time steps of the PC flag





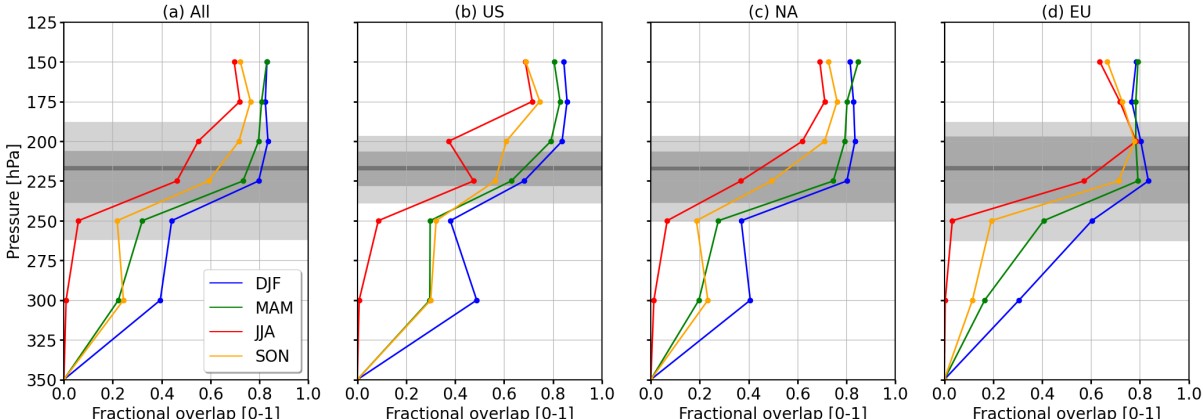

**Figure 3.** Vertical distribution of persistent contrail overlap or intersection $\mathcal{I}$ (unitless) for **(a)** the full domain, as well as **(b)** the US, **(c)** the Atlantic Ocean, and **(d)** the Europe sub-domains, respectively. The color code represents winter (blue), spring (green), summer (red), and autumn (orange). IAGOS-based flight pressure distributions are indicated in gray giving the $10^{th}$, $25^{th}$, $50^{th}$, $75^{th}$, and $90^{th}$ percentiles.

leads to overlap masks between the individual $p$-levels. The overlap mask is one in locations where two adjacent layers allow

PC formation, and is otherwise zero. $\mathcal{I}$ is then calculated between each $p$-level from the ratio of pixels set to 1 in the overlap mask divided by the number of pixels flagged for PC formation from the $p$-level below (higher $p$). The algorithm to calculate $\mathcal{I}$ is propagated upward (from higher to lower $p$-levels) following moisture advection from lower altitudes and $\mathcal{I}$ calculated between 350 and 300 hPa is assigned to the 300 hPa $p$-level and so forth.

First, we calculate $\mathcal{I}$ for the entire domain. Figure 3a shows that $\mathcal{I}$ is subject to a seasonal variation with largest $\mathcal{I}$ of 0.81

(200 hPa) during winter that is followed by spring 0.8 (200 and 150 hPa), autumn 0.75 (175 hPa), and summer 0.72 (175 hPa). The order of $\mathcal{I}$ remains constant over all $p$-levels and follows the seasonality of PC occurrence with highest $\mathcal{P}$ during the winter months. Irrespective of the season, $\mathcal{I}$ increases with decreasing $p$-level, which implies that, if a PC region is present at a certain level $p$, the $p$-level above often contains a PC region, too, which is located at a very similar position in longitude and latitude. Conversely, it is unlikely that a new PC region is formed when there is no existing PC region below. In other words, PC regions

at higher altitudes exist only when there is a PC region present on the level below that acts as some kind of humidity supply, e.g., by atmospheric rivers or convection. On a regional perspective the vertical distributions of $\mathcal{I}$ are similar in shape and magnitude (see Fig. 3b–d). Generally higher values of $\mathcal{I}$ are found for the NA and the EU sub-domain with maxima around 0.8 among the 225 and 200 hPa $p$-level. Maximum $\mathcal{I}$ of the US domain is shifted upward and located between the 200 and 175 hPa $p$-levels.

The vertical distributions of $\mathcal{P}$ and $\mathcal{I}$ imply that today's aircraft fly at altitudes with the highest chance for PC formation, which are also well extended on adjacent layers within the range of pressure levels studied here. This suggests that contrail mitigation by changing flight altitudes might involve large altitude changes.



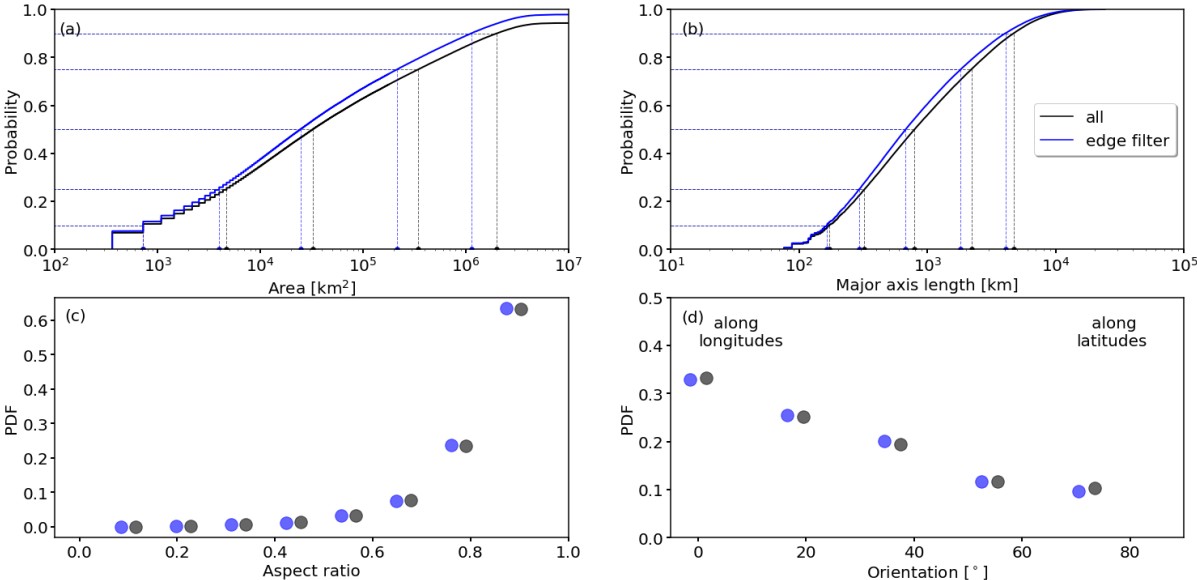

**Figure 4.** Analysis of the morphology of persistent contrail areas over the whole domain. Cumulative distribution functions of **(a)** the area (in km$^2$, top left) and **(b)** the maximum dimension (in km, top right). Normalized probability density functions of **(c)** the aspect ratio $\mathcal{Z}$ (lower left) and **(d)** the orientation of individual persistent contrail formation regions (lower right) for bin sizes of 0.1 and 15$^\circ$, respectively. Pressure levels 250, 225, and 200 hPa are are combined. For each parameter two distributions are given: including (black) and excluding (blue) PC regions that straddle the domain boundary. In **(a)** and **(b)** the 10$^{th}$, 25$^{th}$, 50$^{th}$, 75$^{th}$, and 90$^{th}$ percentiles are indicated by dashed lines.

### 3.3 Size, shape, and orientation of individual PC formation regions derived from ERA5

As described in the previous section, rerouting flights vertically to reduce contrail formation might be impractical. In addi-

tion, changing flight altitude carries the risk to operate aircraft outside their optimal performance envelope and might be often restricted by air traffic control. Alternatively, contrail formation regions could be laterally avoided. In that case, estimates of typical horizontal extent, shape, and orientation of contrail formation regions are important information for rerouting considerations. Those properties are presented in this section with the focus on the radiatively-effective PC regions. The subsequently provided values include all PC regions also the ones that straddle the domain boundaries. In those cases the given sizes represent

a lower estimate of the actual size.

Using the `scikit-image` processing tool (see Section 2.4), the area $\mathcal{A}$, the aspect ratio $\mathcal{Z}$, the orientation angle $\gamma$, and the maximum dimension $D$ of individual PC are identified for each pressure level. ERA5 data from the years 2015 to 2021 is used at its highest temporal and spatial resolution. To limit computational time and to reduce auto-correlation, twelve random, unique days are selected from each month. From each random day, model lead times 0, 6, 12, and 18 UTC are extracted. PC

regions that touch the boundaries of the full domain are kept in the analysis but plotted separately.





First, we discuss the area $\mathcal{A}$ (in unit of km$^2$) of PC regions (see Fig. 4a). In general, $\mathcal{A}$ of individual PC domains are similar on all three $p$-levels between 250 and 200 hPa (see Fig. 4a). This matches with findings from the previous section that PC formation at one level is accompanied by PC formation at neighboring levels. For half (50$^{\text{th}}$ percentile) of the PC regions $\mathcal{A}$ is smaller than 32100 km$^2$ (250 hPa) and 35000 km$^2$ (200 hPa). For illustration, these regions are approximately equivalent to the size of Belgium or Maryland. Considering only the lower 25$^{\text{th}}$ percentile, $\mathcal{A}$ of 4300 km$^2$ (250 hPa) and 5400 km$^2$ (200 hPa) are found. Figure 4a also shows that $\mathcal{A}$ is slightly sensitive to the filtering of edge-straddling PC regions. Ignoring PC regions that interact with the domain boundary primarily removes large domains, which gives more weighting to smaller PC regions.

Similarly, the CDFs of maximum dimension $D$ (in unit of km) are characterized by a steep increase at small $D$ (see Fig. 4b). However, the distributions of $D$ and $\mathcal{A}$ do not directly correlate because PC are often of irregularly shaped. For all three $p$-levels similar distributions of $D$ are derived with 50 % of $D$ being shorter than 760 km (200 hPa) and 820 km (250 hPa). 25 % of the PC regions have $D$ ranging from 310 km (200 hPa) to 330 km (250 hPa). Similar to the distribution of $\mathcal{A}$, $D$ is sensitive to ignoring boundary-straddling PC regions. Doing so primarily ignores PC regions with $D > 1000$ km, which gives more weighting to smaller $D$ and shifts the distribution (blue) to smaller $D$ compared to the CDF including all PC regions (black).

Distributions of the aspect ratio $\mathcal{Z}$ (unitless) are given in Fig. 4c with $\mathcal{Z}$ being similar on the three $p$-levels and being characterized by a steep decline in frequency from $Z = 1$ towards 0.7. This suggests that the majority of PC regions have a 1:1 width-to-length-ratio, while elongated PC regions are less frequent. However, note that larger PC regions tend to be elongated, as discussed later in this section. Filtering for edge-straddling leads to similar distributions in $\mathcal{Z}$.

The orientation of PC formation regions is defined by the angle $\gamma$ (in unit of degrees) between the maximum dimension $D$ and lines of constant latitude. Distributions of $\gamma$ combined for the three $p$-levels are given in Fig. 4d for bins of 15°. In general, all distributions are characterized by a maximum at $\gamma \approx 0°$, indicating the longest axis of majority PC areas is aligned with the West-East direction. Following a West–East alignment is likely a result from the zonally dominated wind field of the mid-latitudes. The distributions of $\gamma$ show that half (50$^{\text{th}}$ percentile) of the PC regions align by $\gamma < 30°$ with the parallels. In these cases a lateral flight diversion would reduce the time spent inside the PC zone with limited additional fuel consumption. For $\gamma >= 30°$ additional fuel consumption is expected to increase. Filtering for edge-straddling leads to similar distributions in $\gamma$, indicating that derived $\gamma$ is relative insensitive to a potential cut-off of the PC regions.

Combining the distributions of $\mathcal{Z}$, the orientation $\gamma$, and the maximum length $D$ provides further insights into the overall appearance of PC regions. Merging the distributions from Fig. 4b and d leads to Fig. 5a, which shows that PC regions with the largest $D$ are dominated by a zonal alignment. Hence, elongated PC regions tend to be aligned along parallels. Figure 5b shows the convoluted distributions from Fig. 4b and c, which indicates that PC regions are generally small and the majority of these regions with $D < 1000$ km tend to a circular shape. Combining the distributions from Fig. 4c and d, leads to the 2D histogram shown in Fig. 5c, which indicates that elongated PC regions are more likely to have an orientation with $\gamma$ close to 0, being aligned along parallels.





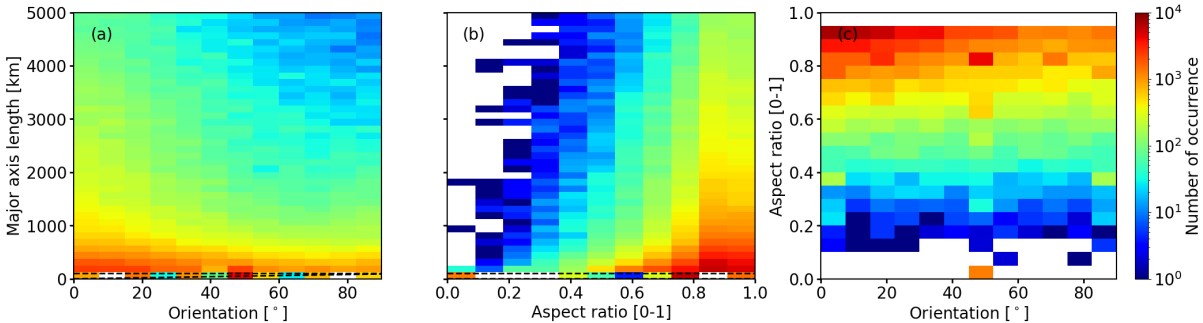

**Figure 5.** Normalized two–dimensional histograms of: **(a)** maximum dimension $D$ (in unit of km) and orientation $\gamma$ (in unit of °); **(b)** maximum dimension and aspect ratio $\mathcal{Z}$ (unitless); and **(c)** $\mathcal{Z}$ and orientation $\gamma$. The number of occurrence is color-coded on a logarithmic scale.

### 3.4 Climatologies of temperature, relative humidity, and persistent contrail formation potential.

PC occurrence is mainly driven by the temporal and spatial distribution of $T_{\mathrm{ERA}}$ and $r_{\mathrm{ERA,ice}}$, which manifest itself in the
seasonal variability of $\mathcal{P}$. To better understand the distributions of $\mathcal{P}$ that were presented in Section 3.2, we calculate and provide climatologies of PC formation in relation to climatologies of ambient conditions. In other words, we answer the question: How does the temporal-spatial distribution of $T_{\mathrm{ERA}}$ and $r_{\mathrm{ERA,ice}}$ prescribe the distribution of PC formation and $\mathcal{P}$? The investigated domain includes the full domain that was defined in Section 2.1.

Climatologies of $T_{\mathrm{ERA}}$, $r_{\mathrm{ERA,ice}}$, wind speed $U_{\mathrm{ERA}}$, and the PC formation potential $\mathcal{P}$ are calculated from the years 2015 to
2021 (included). To reduce auto correlation and to limit computational time, ERA5 data was aggregated by extracting $T_{\mathrm{ERA}}$, $r_{\mathrm{ERA,ice}}$, and $U_{\mathrm{ERA}}$ for each day of the month at lead times of 0, 6, 12, and 18 UTC only. The spatial resolution was reduced by selecting every second grid box, i.e., every 0.5°. Then the data was spatially and temporally averaged depending on zonal or temporal averaging. The PC occurrence was estimated with the SAc on the basis of the extracted $T_{\mathrm{ERA}}$ and $r_{\mathrm{ERA,ice}}$ fields, and averaged afterwards. $T_{\mathrm{ERA}}$ and $r_{\mathrm{ERA,ice}}$ are selected as they are the primary input for the SAc. Information about $U_{\mathrm{ERA}}$ is
added to infer locations of high wind speeds, i.e., the jet stream, which is suspected to support contrail formation (Irvine et al., 2012). It is hypothesized that the curvature of the jet stream as well as wind shear along the jet stream trigger the advection and adiabatic cooling of air from lower altitudes, which promotes contrail formation. Further more, the jet stream is an atmospheric feature that is frequently used by aircraft on transatlantic flights, which makes it interesting in relation to PC regions.

First, climatologies of zonally-averaged, vertical cross-sections of $T_{\mathrm{ERA}}$ are discussed (first column in Fig. 6). Irrespective of
the season, lines of constant temperature (isotherms) are slanted such that at constant $p$-levels, temperatures decrease poleward. The pattern of $T_{\mathrm{ERA}}$ is controlled by the differential heating of the Earth surface and a near-surface, zonal temperature gradient. The temperature gradient is counter-acted by large-scale circulations, i.e., the Hadley cell and the polar cell, which lead to a net energy transport from the Equator towards the poles. The energy surplus at the surface also propagates to higher altitudes. However, the lowest temperatures are found closest to the Equator (at 30°N ) at 150 hPa, creating a strong vertical temperature







**Figure 6.** From left to right: Monthly mean temperature of $T_{\text{ERA}}$ (in unit of K), relative humidity $r_{\text{ERA,ice}}$ (in unit of %), wind speed $U_{\text{ERA}}$ (in unit of m s$^{-1}$), and contrail formation potential $\mathcal{P}$ as a function of latitude and pressure level $p$. The right most column shows the vertical flight pressure distribution (FPD). From top to bottom: Climatologies for winter (DJF), spring (MAM), summer (JJA), and autumn (SON).

gradient that is indicated by the narrow isotherms. The strongest vertical temperature gradient $\Delta T$ between the 350 and 150 hPa $p$-level is calculated for the summer months with $\Delta T = 34$ K (211 to 239 K) and the smallest for the winter months ($\Delta T = 22$ K). The stratiform pattern of the isotherms and the gradient (at 30°N) is broken up north of 40°N and for $p < 250$ hPa. In other words, the distribution of $T_{\text{ERA}}$ becomes less sensitive to the latitude and the $p$-level at high latitudes and low pressures.

The Hadley cell and the polar cell also influence the distribution of relative humidity. The resulting climatologies of $r_{\text{ERA,ice}}$

are shown in the second column in Fig. 6. For all seasons, the highest $r_{\text{ERA,ice}}$ are found at 350 hPa close to 60°N. However, during summer the region with $r_{\text{ERA,ice}} > 60$ % propagates further to the south (from 45°N to 60°N) and to lower $p$-levels (300 to the 250 hPa $p$-level). In spring and summer, intermediate values are determined. Regions of zonal averages with the highest




$r_{\mathrm{ERA,ice}}$ are enclosed by drier air from above and below. At a first glance, the widespread occurrence of $r_{\mathrm{ERA,ice}} > 60\%$ in summer contradicts the vertical distributions of PC that were shown in Fig. 6, with PC being least frequent in summer. But

recall that temperature is also important to fulfill the SAc: While $r_{\mathrm{ERA,ice}}$ might be sufficient for PC formation, $T_{\mathrm{ERA}}$ is above $T_{\mathrm{crit}}$ and, therefore, no PC formation is possible.

The resulting distributions of PC formation are constrained by $T_{\mathrm{ERA}}$ and $r_{\mathrm{ERA,ice}}$ twofold by: (i) cold but dry air from aloft and (ii) humid but too warm air from below. PC formation can only take place in a narrow range, where all criteria for PC formation are met (see third column in Fig. 6). The resulting distributions of PC formation are slanted from high (30°N)

to lower $p$-levels (60°N) and follow the isotherms and lines of equal $r_{\mathrm{ERA,ice}}$ (isohumes). The region with $\mathcal{P} > 0.05$ has the largest vertical extent during winter (Fig. 6d) and is thinnest in summer (Fig. 6s). The thinning during the summer months is a results of the strong gradients in $T_{\mathrm{ERA}}$ and $r_{\mathrm{ERA,ice}}$, which narrow the potential PC formation region by high $T_{\mathrm{ERA}}$ below and low $r_{\mathrm{ERA,ice}}$ from above. The resulting distributions of PC formation in Fig. 6 (fourth column), with the highest chance and extension for PC formation in winter and the lowest in summer, look like the vertical distributions of PC given in Fig. 2a.

Column 5 in Fig. 6 provides information about the flight pressure distributions (FPDs) derived from IAGOS observations representing commercial, transatlantic flights. Irrespective of the seasons the majority of aircraft fly at similar altitude / $p$-level, where the maximum PC formation potential is identified. To minimize the chance for PC formation, the overlap of PC occurrence and FPD must be minimized, for example, by shifting the average flight altitude upwards (lower $p$-levels). Due to the slanted lines of equal PC formation potential, the required shift in flight altitude is smaller at 60°N than at 30°N. It is

noted that close to 60°N the transition to higher altitude / lower $p$-levels might be associated with more flights in the lower stratosphere, where the climate impact of non-$CO_2$ effects, including emission of water vapor and nitrogen oxides, is enhanced.

Eastward flight trajectories across the Atlantic regularly take advantage of the jet stream to reduce fuel consumption and flight time. Highest wind speeds $U_{\mathrm{ERA}}$ are identified between $p$-levels 300 and 200 hPa and latitudes of 45°N to 55°N. $U_{\mathrm{ERA}}$ is subject to seasonal variations with the highest mean $U_{\mathrm{ERA}}$ during winter with $U_{\mathrm{ERA}} > 35\ \mathrm{m\,s^{-1}}$ and a weaker jet stream in

summer with $U_{\mathrm{ERA}} > 25\ \mathrm{m\,s^{-1}}$. In addition, the region of highest $U_{\mathrm{ERA}}$ is shifted further to the North during summer, which is a result of the northward shift of the Hadley and polar cell. Intermediate values of $U_{\mathrm{ERA}}$ are determined for spring and autumn, which act as transition periods. Similar seasonal variation in the strength and location of maximal $U_{\mathrm{ERA}}$ were also identified by, for example, Pena-Ortiz et al. (2013) and Hall et al. (2015).

Regions of large PC occurrence seem correlated with regions of high $U_{\mathrm{ERA}}$ (Columns 3 and 4 of Fig. 6). However, it is

unclear to the authors whether this is a coincidence, i.e., the meteorological conditions favor both the jet stream position and PC formation, or if the jet stream itself promotes PC formation by bringing humid air aloft. A meandering jet stream that changes its speed and direction, e.g., strong curvature in a trough, might trigger advection of air from below because of mass conservation (Riehl et al., 1952; Beebe and Bates, 1955; Nakamura, 1993). As described earlier, the lifted air is adiabatically cooled and moistened, which supports PC formation. In any case, the position of highest wind speed might be used as a proxy

for potential PC occurrence. In this case the avoidance of the jet stream on westbound flights (minimize head wind) goes hand in hand with PC formation mitigation, while on eastbound flights (take advantage of tailwind), aircraft navigate in a regime with the highest formation potential. However, in these cases fuel consumption and emitted WV are reduced and the trade-off





between flying within or outside the jet stream would have to be quantified for each flight. Furthermore, eastbound flights are mostly at night and, hence, induce a positive net warming.

## 4 Summary and discussion

This study applies the SAc and persistence criterion to distributions of $T$ and $r$ from ERA5 to study the large-scale distribution and morphology of regions of PC formation. The fitness for purpose of ERA5 for this kind of evaluation was demonstrated by Wolf et al. (2023a). The analysis in the present study focused on the North Atlantic flight corridor spanning from the East coast of North America (110°W) to central Europe (30°E) and between 30°N and 70°N, and seven years of data from 2015 to 2021. Here we presented distributions of crossing length $\mathcal{L}$ - the distance an aircraft crosses a PC region - based on ERA5 data and IAGOS observations. Median $\mathcal{L}$ values of 9 and 66 km were found using the native and the time-averaged IAGOS data set, respectively. The time-averaged version was introduced to mimic the average spatial resolution of ERA5 at 19 km. For ERA5 a median $\mathcal{L}$ of 155 km was identified. The differences in $\mathcal{L}$ between the time-averaged IAGOS data set and ERA5 are explained by the higher natural variability in relative humidity in the IAGOS observations compared to the ERA5 grid box mean value.

The morphology of individual PC formation regions was determined by applying the python image processing tool `scikit-image` (van der Walt et al., 2014) to the 2D binary arrays of PC occurrence. To the authors knowledge, this is the first time contrail formation is looked at in this way. The build-in functions of `scikit-image` provide the surface area $\mathcal{A}$, the maximum dimension $D$, and the orientation of individual detected PC regions. The PC regions that straddle the boundaries of the domain are identified because in those cases the PC dimensions are underestimated. 50 % of the identified PC regions were smaller than 32000 km² (250 hPa) and 35000 km² (200 hPa). A general increase in $\mathcal{A}$ with decreasing $p$-level (250–200 hPa) was found due to a colder but still moist enough atmosphere. Median maximum dimension $D$ of 760 km (200 hPa) and 820 km (250 hPa) were found. Both, $\mathcal{A}$ and $D$, are slightly sensitive to the inclusion of PC regions that straddle the domain boundaries because larger PC regions are most likely to be straddling. The orientation of PC regions was specified by the angle $\gamma$ between the major axis length (major extension) with respect to parallels. Therefore, PC regions have a tendency to align along lines of constant latitude ($\gamma = 0°$) with a decreasing probability of occurrence with increasing $\gamma$. This indicates that PC regions preferably align within the dominant westerly flow that is present at high altitudes, e.g., the jet stream. Analysis of the aspect ratio $\mathcal{Z}$, which is the ratio of major to minor axis length, indicates that PC regions are mostly of near-circular shape or slightly oval with $\mathcal{Z}$ down to 0.7, while $\mathcal{Z} < 0.7$ are rare. Larger PC regions are more likely to be elongated. The stretching along one dimension likely results from being embedded in the westerly flow.

Seasonal, vertical distributions of the PC formation potential $\mathcal{P}$ indicate maximum $\mathcal{P}$ are found on $p$-levels 250, 225, and 200 hPa, where most of the air traffic takes place. Vertical variations of $\mathcal{P}$ were identified among the three sub-domains with a decrease in altitude from West to East. Furthermore, the magnitude of $\mathcal{P}$ was found to be sensitive to seasonal variations with lowest $\mathcal{P}$ during summer and highest $\mathcal{P}$ in winter.

In this context, the fractional overlap $\mathcal{I}$ of adjacent $p$-levels for coinciding PC formation regions was investigated. The analysis showed that $\mathcal{I}$ increases with altitude indicating that existing PC regions overlap. However, the total size $\mathcal{A}$ of PC




regions decreases with altitude. Consequently, PC formation regions, if present, penetrate multiple $p$-levels and overlap, instead of being horizontally displaced and separated on adjacent $p$-levels. This suggests that vertical contrail avoidance will in many cases involve large altitude changes.

Finally, climatologies of $T_{\mathrm{ERA}}$, $r_{\mathrm{ERA,ice}}$, wind speed $U_{\mathrm{ERA}}$, and related PC formation potential $\mathcal{P}$ were presented. These 450 climatologies characterize the temporal and spatial distribution of PC regions depending on the ambient conditions. Vertical cross-sections ($p$-Latitude) of climatologies of $\mathcal{P}$ showed largest vertical extend during winter months, while the vertical extend is smallest in summer. The vertical extension of PC formation regions and related $\mathcal{P}$ in summer is restricted by too high temperatures from below ($p > 250$ hPa) and dry air masses from above ($p < 200$ hPa), which both inhibit PC formation. In addition, the overall magnitude of $\mathcal{P}$ is generally lower in summer than in winter. The climatologies of $T_{\mathrm{ERA}}$ and $r_{\mathrm{ERA,ice}}$ 455 revealed a slanted alignment of isotherms (lines of constant temperature) and isohumes (lines of constant moisture). This is expected from the differential heating of the Earth surface. The slanted distribution of temperature and relative humidity propagated into a slanted distribution of $\mathcal{P}$, with lines of constant $\mathcal{P}$ decreasing in altitude / increasing in $p$-level from 30°N to 60°N. This implies that larger altitude changes are required in the mid-latitudes compared to polar regions. The analysis further suggested that enhanced values of $\mathcal{P}$ and high wind speeds are co-located. Consequently, the jet stream is a region where PC 460 formation regions may be difficult to avoid.

*Code availability.* The python code that was used to perform the analysis and the quantile correction is provided following: https://doi.org/10.5281/zenodo.8418565

*Data availability.* ERA5 data can be obtained from the European Centre for Medium-Range Weather Forecasts (ECMWF) data catalog at https://doi.org/10.24381/cds.f17050d7 (Hersbach et al., 2023).
465 The IAGOS data can be downloaded from the IAGOS data portal at https://doi.org/10.25326/20 (Boulanger et al., 2020).

*Author contributions.* **KW** performed the data analysis and prepared the manuscript. **NB** and **OB** contributed equally to the preparation of the manuscript.

*Competing interests.* The authors declare no competing interest.

*Acknowledgements.* This research has been supported by the French Ministère de la Transition écologique et Solidaire (N° DGAC 382 470 N2021-39), with support from France's Plan National de Relance et de Resilience (PNRR) and the European Union's NextGenerationEU.



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
