# Peer review of "Distribution and morphology of non-persistent and persistent contrail formation areas in ERA5."

_EGUsphere, 2023_

## Referee Comment (RC1)

Review of "Distribution and morphology of non-persistent and persistent contrail formation areas in ERA5", by Wolf, Bellouin and Boucher, egusphere-2023-3086

This is a most impressive, comprehensive article, characterizing the potential persistent contrail (PC) formation conditions as a function of time of year, temperature, relative humidity, pressure, and wind speed. The region considered is from the North Atlantic flight corridor from the East coast of North America to central Europe and between 30°N and 70°N, using data from 2015 to 2021. The modified Schmidt-Appleman criteria are used to identify the PC regions, using a combination of In-service Aircraft for a Global Observing System (IAGOS) data and ERA5 re-analysis products. Most interestingly, the dimensions of individual PC formation regions was determined by applying the python image processing tool scikit-image Python.

Some takeaways that I got from the article. Most commercial aircraft are currently flying at altitudes that are most prone to PC formation, thus, shifting to probably lower altitudes would decrease PC formation, but this is not practical. Also, that the position of highest wind speed might be used as a proxy for potential PC occurrence. It's interesting, using the angle between the elongated PC regions and latitude that lateral flight diversion would reduce the time spent inside the PC zone with limited additional fuel consumption.

Some quality controls were used in the analysis. A 2023 study noted by Wolf et al. evaluated the IAGOS observations, with directly measured temperature and relative humidity with a Vaisala sensor from commercial aircraft, to evaluate ERA5 performance. They found a dry bias and applied a correction to those data.

My comments appear below. I recommend publication of the article subject to minor revision.

**Main Comments**

The Vaisala Humicap sensor is used for the relative humidity measurements for IAGOS. I assume the sensor is similar to a Vaisala RS80H (Humicap) sensor. A study by Verver et al. (2006) found a significant wet bias in RH from +2% to +5% in the RS80H profiles in the upper troposphere when compared with a very advanced humidity sensor. Please comment on this as it may have effected the dry bias found in the ERA5 data.

Verver, G., M. Fujiwara, P. Dolmans, C. Becker, P. Fortuin, and L. Miloshevich, 2006: Performance of the Vaisala RS80A/H and RS90 Humicap Sensors and the Meteolabor "Snow White" Chilled-Mirror Hygrometer in Paramaribo, Suriname. *J. Atmos. Oceanic Technol.*, **23**, 1506–1518

Line 143. How are the time-averages calculated? Is it from the current point to 19.4 seconds ahead of the aIrcraft?

Do NPC satisfy the SAc criteria but not rh>rhi? I assume that's the case but I suggest stating it.

A schematic showing the different path lengths for IAGOS and EROS would be helpful, and showing the distances for time-averaged values.

Figure 6. This is a most interesting and important figure.

Years ago, there was a program called GHOST (Global HOrizontal Sounding Technique), which used constant pressure balloons to measure ambient temperature and relative humidity. Also, more recently, the CNES new super-pressure balloon system deployed for Strateole-2 program. Would such systems potentially aid in evaluating the ERA5 data in the future?

Minor Comments

138. "fixed grid resolution" is repeated in this line

167. allows us

249: could you define "all grid boxes"

250. for pressure
362. Further more>Furthermore

---

## Author Comment (AC1)

**Reply to Reviewer #1**
(Referee comment on "Distribution and morphology of non-persistent and persistent contrail formation areas in ERA5" by K. Wolf et al. (egusphere-2023-3086), https://doi.org/10.5194/egusphere-2023-3086-RC1, 2024)

We thank the Reviewer for the time she/he spent on the manuscript, and for the useful comments. Addressing the comments has improved the manuscript.

For better legibility, the Reviewer's comments are highlighted in **bold** and changes in the manuscript are in *italics*.
* * *
**This is a most impressive, comprehensive article, characterizing the potential persistent contrail (PC) formation conditions as a function of time of year, temperature, relative humidity, pressure, and wind speed. The region considered is from the North Atlantic flight corridor from the East coast of North America to central Europe and between 30◦N and 70◦N, using data from 2015 to 2021. The modified Schmidt-Appleman criteria are used to identify the PC regions, using a combination of In-service Aircraft for a Global Observing System (IAGOS) data and ERA5 re-analysis products. Most interestingly, the dimensions of individual PC formation regions was determined by applying the python image processing tool scikit-image Python. Some takeaways that I got from the article. Most commercial aircraft are currently flying at altitudes that are most prone to PC formation, thus, shifting to probably lower altitudes would decrease PC formation, but this is not practical. Also, that the position of highest wind speed might be used as a proxy for potential PC occurrence. It's interesting, using the angle between the elongated PC regions and latitude that lateral flight diversion would reduce the time spent inside the PC zone with limited additional fuel consumption.**
**Some quality controls were used in the analysis. A 2023 study noted by Wolf et al. evaluated the IAGOS observations, with directly measured temperature and relative humidity with a Vaisala sensor from commercial aircraft, to evaluate ERA5 performance. They found a dry bias and applied a correction to those data. My comments appear below. I recommend publication of the article subject to minor revision.**

We thank the Reviewer for the positive appreciation of our manuscript.

**Main Comments**
**The Vaisala Humicap sensor is used for the relative humidity measurements for IAGOS. I assume the sensor is similar to a Vaisala RS80H (Humicap) sensor. A study by Verver et al. (2006) found a significant wet bias in RH from +2% to +5% in the RS80H profiles in the upper troposphere when compared with a very advanced humidity sensor. Please comment on this as it may have effected the dry bias found in the ERA5 data. Verver, G., M. Fujiwara, P. Dolmans, C. Becker, P. Fortuin, and L. Miloshevich, 2006: Performance of the Vaisala RS80A/H and RS90 Humicap Sensors and the Meteolabor "Snow White" Chilled-Mirror Hygrometer in Paramaribo, Suriname. J. Atmos. Oceanic Technol., 23, 1506–1518**

Verver et al. (2006) investigated the performance of several Vaisala radiosondes, among others, of types RS80A, RS80H, and RS90H. Those were compared with a chilled-mirror hygrometer. RS80H and RS90H are both equipped with a Vaisvala Humicap-H sensor that

the Reviewer is referring to.

Verver et al. (2006) did not make any specific remarks about biases above 7 km altitude, which makes it difficult to compare with IAGOS measurements. Verver et al. (2006) only stated a good correlation between the radiosonde observations and the chilled-mirror hygrometer. Radiosonde measurements further require corrections to consider for, among other factors, for sensor time lag / response time and potential insolation (e.g., Miloshevich et al. (2004) and Heymsfield et al. (1998)). These can cause potential biases. In addition, transferring the measurement performance of radiosonde equipment to aircraft measurements is difficult as they operate under different technical constraints and operational conditions. The corrections for aircraft measurements of relative humidity are different as thermodynamic effects, due to the high wind velocity around the sensor inlet, have to be considered (e.g., Neis et al. (2015) and Petzold et al. (2017)).

However, we acknowledge that IAGOS might be subject to a humidity bias of some kind. In this regard we refer to the comparison study by Petzold et al. ( 2020), who showed good statistical agreement between IAGOS measurements and dedicated humidity measurements.

**Line 143. How are the time-averages calculated? Is it from the current point to 19.4 seconds ahead of the aircraft?**

To be clearer this section has been extended and rephrased to the following:

*"The IAGOS measurements are averaged by applying a Gaussian filter. The standard deviation σ of the Gaussian filter is approximated with:*
*σ = (k − 1) / 6,*
*where k is the window length of the smoothing filter. To average over 19 km, we set σ = 3, based on an assumed average cruise speed of around 240 m s⁻¹ and a resulting segment length (distance between two measurements) of around 1 km."*

**Do NPC satisfy the SAc criteria but not rh>rhi? I assume that's the case but I suggest stating it.**

We followed the suggestion of the Reviewer and made the paragraph clearer.

*"Within this study we use the revised version of the SAc following Schumann (1996) and Rap et al. (2010). General details on the SAc and equations required to calculate $T_{crit}$ and $r_{crit}$ can be found in Rap et al. (2010) or Wolf et al. (2023a). Within the present study the same definitions and nomenclature as in Wolf et al. (2023a) are used, and data points are categorized for non-persistent contrails (NPC), persistent contrails (PC), and reservoir (R) conditions. Data points that are flagged for NPC fulfill the SAc, but the ambient air is sub-saturated with respect to ice (100 % < rice). Samples that are flagged for PC fulfill the SAc and are saturated with respect to ice ($r_{ice}$ > 100 %). Data points that are flagged for reservoir conditions fulfill the criteria for ice-supersaturation but fail the SAc. Discussion on the Reservoir category can be found in Wolf et al. (2023a). All data points that are not assigned to one of the groups are labeled as non-contrail (NoC)."*

**A schematic showing the different path lengths for IAGOS and EROS would be helpful, and showing the distances for time-averaged values.**

We appreciate the Reviewer's comment and we would like to make the paper more readable. However, it is not clear to us what the Reviewer wants to be displayed, i.e., what she / he means with different path lengths / segment length. The paths length is a distance

and the distribution of crossing length for NPC, PC, and reservoir conditions are presented in a Fig.1.

**Figure 6. This is a most interesting and important figure. Years ago, there was a program called GHOST (Global HOrizontal Sounding Technique), which used constant pressure balloons to measure ambient temperature and relative humidity. Also, more recently, the CNES new super pressure balloon system deployed for Strateole-2 program. Would such systems potentially aid in evaluating the ERA5 data in the future?**

Yes, additional in-situ measurements by balloons would be beneficial, particularly when the observations cover a longer time period like it seems to be the case for the Strateole-2 program. In-situ observations in the upper troposphere and lower stratosphere are sparse. Available measurements typically stem from dedicated measurement campaigns that target specific regions or atmospheric features for a limited period of time. Remote sensing observations, e.g., from satellite or ground, rely on assumptions for their retrieval products that introduce uncertainties. However, the sampling by individual balloons might be too limited to provide a strong constraint on numerical weather forecasting systems.

**Minor Comments**

**138. "fixed grid resolution" is repeated in this line**
The doubled part of the sentence has been removed.

**167. allows us**
"us" has been added to the sentence.

**249: could you define "all grid boxes"**

To be more specific, the sentence has been rephrased to the following:

*"P is calculated for each p-level as the ratio of PC flagged grid-boxes in relation to the total number of grid boxes in the investigated domain and is then averaged over time steps and months."*

**250. for pressure**

The Reviewer is right and the word was exchanged:

*"For p-levels below 225 hPa the distributions are dispersed suggesting a larger seasonal variability. "*

**362. Further more>Furthermore**
The typo has been corrected.

Heymsfield, A. J. / Miloshevich, L. M. / Twohy, C. / Sachse, G. / Oltmans, S., Upper-tropospheric relative humidity observations and implications for cirrus ice nucleation, 1998, Geophys. Res. Let. , Vol. 25, No. 9, p. 1343-1346

Miloshevich, L. M. / Paukkunen, A. / Vömel, H. / Oltmans, S. J., Development and Validation of a Time-Lag Correction for Vaisala Radiosonde Humidity Measurements, 2004, J. Atmos. Ocean. Tech. , Vol. 21, No. 9, American Meteorological Society: Boston MA, USA, p. 1305-1327

Neis, P. / Smit, H. G. J. / Rohs, S. / Bundke, U. / Krämer, M. / Spelten, N. / Ebert, V. / Buchholz, B. / Thomas, K. / Petzold, A., Quality assessment of MOZAIC and IAGOS capacitive hygrometers: insights from airborne field studies, 2015, Tellus B: Chem. Phys. Meteorol. , Vol. 67, No. 1, p. 28320

Petzold, A. / Neis, P. / Rütimann, M. / Rohs, S. / Berkes, F. / Smit, H. G. J. / Krämer, M. / Spelten, N. / Spichtinger, P. / Nédélec, P. / Wahner, A., Ice-supersaturated air masses in the northern mid-latitudes from regular in situ observations by passenger aircraft: vertical distribution, seasonality and tropospheric fingerprint, 2020,Atmos. Chem. Phys. , Vol. 20, No. 13, p. 8157-8179

---

## Author Comment (AC2)

**Reply to Reviewer #2**

(Referee comment on "Distribution and morphology of non-persistent and persistent contrail formation areas in ERA5" by K. Wolf et al. (egusphere-2023-3086), https://doi.org/10.5194/egusphere-2023-3086-RC2, 2024)

We thank the Reviewer for the time she/he spent on the manuscript, and for the useful comments. Addressing the comments has improved the manuscript.

For better legibility, the Reviewer's comments are highlighted in **bold** and changes in the manuscript are in *italics*.
* * *
**Review of "Distribution and morphology of non-persistent and persistent contrail formation areas in ERA5" by Wolf et al., egusphere-2023-3086**

**This paper describes an analysis of the characteristics and morphology of contrail formation regions from the IAGOS aircraft data and the ERA5 reanalysis. The focus is on the North Atlantic flight track around cruise altitudes. Characterising regions of persistent contrail formation is important and very relevant for the potential mitigation of the climate impact of aircraft flights by rerouting. There are a number of interesting results from the analysis, with an overall conclusion that some persistent contrail regions will be difficult to avoid by rerouting aircraft because of their large vertical and horizontal extents and their frequent colocation with the jet stream.**

**The paper is well written and logically presented and is appropriate for ACP. I have one main comment and several minor corrections that need to be addressed.**

We thank the Reviewer for the positive appreciation of our manuscript.

**MAIN COMMENT**

**Should the crossing length for ERA5 in Figure 1 and Table 2 have values less than the grid length of 14km - 24km (i.e. values in Fig 1 for ERA5 go down to less than 1km - the native IAGOS resolution - and Table 2 has a 10th percentile for ERA5 PC of 9km)? Line 211 says "short crossing lengths occur less frequently and cannot by construction be smaller than grid-box size". I assume the reason for small lengths is because the flight track can cut the corner of a grid cell, but then the above sentence is not correct and the reason should be made clear in the text. Whether this makes sense is another matter, as the ERA5 data is not providing useful information less than the grid length (and probably several grid lengths). Would a different methodology (e.g. a flight track at the resolution of the grid) lead to a significant difference in the results/conclusions?**

The Reviewer addresses an important point.
In the version of the manuscript that the Reviewer reviewed, we used IAGOS measurements with their native temporal resolution of four seconds. The distance (segment length) between two IAGOS measurements has been approximated as the great circle distance. As a result, the smallest possible distance that could appear in the contrail length statistics is therefore around one kilometer – both in the original IAGOS data and in the collocated ERA5 data. One can interpret this as an aircraft-centered perspective. To

address the Reviewer's comment, we now follow two approaches In the revised manuscript:

(i) An aircraft-centered approach where IAGOS measurements are averaged over 1-km segments. As the Reviewer correctly points out, that poses problems in cases where the IAGOS flight track crosses a contrail-forming region tangentially. So that might not be a fair comparison.

(ii) A model-centered approach, where segments are 19-km long. Here, the limitation is the suppression of contrail features that are smaller than this distance The length of 19 km has been selected according to the average Cartesian grid box resolution of ERA5 (0.25°) in the investigated latitude band. The data is up-scaled and a 19-km segment is flagged for NPC, PC, R, or NoC based on the flag that occurs most frequently over the 19 km section. As a result, the minimum path length is 19 km. Potential smaller-scale contrail features are now down-weighted and suppressed. Based on the updated plot, we revised the section of the text. Due to the extent of the revisions, we  direct the Reviewer to the revised manuscript and the track-changes document.

**MINOR CORRECTIONS**

**Section 2.2 Regarding interpolation of the ERA5 data, it should be noted here that there is already some interpolation from the native grid of ERA5 (reduced gaussian TL639 grid, ~31km approx equally spaced grid, and 137 hybrid sigma-pressure coordinate levels) to the lat/lon 0.25 degree grid and fixed pressure levels. e.g. https://confluence.ecmwf.int/display/CKB/ERA5%3A+What+is+the+spatial+reference**

We agree with the Reviewer that this is an important point. We now mention that ERA5 is a spectral model that internally operates at an approximate resolution of 31 km. The text has been modified as following:

*" [...] The ERA5 data set was generated with the ECMWF Integrated Forecasting System (IFS) cycle Cy41r2 (operational in 2016). ERA5 is a spectral model with an internal resolution of approximately 31 km. Therefore, the HRES product on the 0.25◦ Cartesian grid represents interpolated values from the coarser internal Gaussian grid (Hersbach et al. (2020))."*

**Line 138 Duplication of "fixed grid resolution"**

The second instance has been removed.

**Line 147 bellow → below**

The typo has been corrected.

**Line 156 Given the "reservoir (R)" conditions are mentioned, it is worth just mentioning briefly in one sentence what this refers to, as it may not be obvious to the reader and shouldn't require looking up in a different paper.**

Following this comment, we added a sentence to text which now reads as follows:

*"Within this study we use the revised version of the SAc following Schumann (1996) and Rap et al. (2010). General details on the SAc and equations required to calculate Tcrit and rcrit can be found in Rap et al. (2010) or Wolf et al. (2023a). Within the present study the same definitions and nomenclature as in Wolf et al. (2023a) are used, and data points are*

*categorized for non-persistent contrails (NPC), persistent contrails (PC), and reservoir (R) conditions. Data points that are flagged for NPC fulfill the SAc, but the ambient air is sub-saturated with respect to ice (100 % < rice). Samples that are flagged for PC fulfill the SAc and are saturated with respect to ice (rice > 100 %). Data points that are flagged for reservoir conditions fulfill the criteria for ice-supersaturation but fail the SAc. Discussion on the Reservoir category can be found in Wolf et al. (2023a). All data points that are not assigned to one of the groups are labeled as non-contrail (NoC)."*

**Line 340 relative -> relatively**

The typo was corrected.

**Line 344 "Figure 5b….indicates that PC regions are generally small…" This part of the sentence is a bit vague and the size of PC regions is already quantified in Fig 4b, so I suggest removing it.**

We followed the suggestion of the Reviewer and removed this part of the sentence.

**Line 426 "…to the 2D binary arrays of PC occurrence." I suggest adding "in the ERA5 dataset." to this sentence for clarity.**

Following the suggestion of the Reviewer the proposed wording has been added to the text.

**Lines 451 and 452 extend -> extent**

The typo was corrected.

**Line 455 "isohumes (lines of constant moisture)" should be "(lines of constant relative humidity)" for clarity.**

Thank you for pointing this out. The sentence has been rephrased according to the suggestion.

**Fig 4. Top right x-axis says "Major axis length" but the caption says "maximum dimension" which is 2x the major axis length.**

The Reviewer is right and we have corrected the plot accordingly. The correct labeling is "Maximum dimension" as it is given in the figure caption and in the text.

**Fig 4. caption 4th line "are are"**

The second "are" has been removed.

**Fig 5. Left axis label says "Major axis length" but the caption says "maximum dimension" which is 2x the major axis length.**

The Reviewer is right and we have corrected the plot accordingly. The correct labeling is "Maximum dimension" as it is given in the figure caption and in the text.